# Sexual and Reproductive Health Service Needs Associated with Underage Initiation of Selling Sex among Adult Female Sex Workers in Guinea-Bissau

**DOI:** 10.3390/ijerph191912715

**Published:** 2022-10-05

**Authors:** Ashley Grosso, Lindsay Berg, Katherine Rucinski, Amrita Rao, Mamadú Aliu Djaló, Daouda Diouf, Stefan Baral

**Affiliations:** 1Department of Urban-Global Public Health, Rutgers School of Public Health, Newark, NJ 07102, USA; 2Center for Population Behavioral Health, Rutgers Institute for Health, Health Care Policy and Aging Research, New Brunswick, NJ 08901, USA; 3Department of International Health, Johns Hopkins Bloomberg School of Public Health, Baltimore, MD 21205, USA; 4Department of Epidemiology, Johns Hopkins Bloomberg School of Public Health, Baltimore, MD 21205, USA; 5Enda Santé, Guiné-Bissau, Bissau 1041, Guinea-Bissau; 6Enda Santé, Dakar 3370, Senegal

**Keywords:** female sex workers, Guinea-Bissau, minors, adolescents, reproductive health, sexual health

## Abstract

Objective: To assess the prevalence and predictors of underage initiation of selling sex among female sex workers (FSW) in Guinea-Bissau. Methods: 505 adult FSW recruited using respondent-driven sampling were surveyed in 2017. Multivariable logistic regression was used to identify demographic, behavioral, and psychosocial characteristics associated with initiation of selling sex while underage (<18 years). Results: A total of 26.3% (133/505) of FSW started selling sex before age 18. Underage initiation of selling sex was associated with experiencing forced sex before age 18 (adjusted odds ratio (aOR): 6.74; 95% confidence interval (CI): 2.05–22.13), and never being tested for HIV (aOR: 0.43; 95% CI: 0.20–0.91). Despite having lower odds of wanting to have children or more children (aOR: 0.31; 95% CI: 0.17–0.56), FSW who started selling sex while underage had lower odds of using highly effective contraception such as implants (aOR: 0.43; 95% CI: 0.24–0.77). Among those who were ever pregnant, a lower percentage of FSW who started selling sex while underage accessed antenatal care (56.6% vs. 74.7%, *p* = 0.008). Conclusions: These data suggest that early initiation of selling sex among adult FSW in Guinea-Bissau is common. Social services for youth and integrated HIV and reproductive health services are critical to address the persisting sexual and reproductive health needs of FSW who started selling sex while underage.

## 1. Introduction

The third Sustainable Development Goal on good health and well-being includes targets to end the AIDS epidemic and ensure universal access to sexual and reproductive healthcare services including family planning, information and education by 2030. The fifth Sustainable Development Goal on gender equality also includes a target to ensure universal access to sexual and reproductive health and reproductive rights. Identifying and addressing the needs of vulnerable populations are important components of achieving these targets [1]. Globally, selling sex under the age of 18 is prevalent and confers immediate and long-term vulnerabilities to adverse sexual and reproductive health outcomes [2]. Much of the research on underage entry into selling sex has been conducted in Asia [3,4,5,6,7], Eastern Europe [8,9], and the Americas [10,11,12,13,14,15,16,17,18]. A growing body of studies in Africa have included underage participants who sell sex [19,20,21] or reported the prevalence and predictors of underage entry into selling sex among adult female sex workers (FSW) [22,23,24,25,26,27,28]. Research on this topic is particularly important in Africa due to the disproportionate burden of poor sexual and reproductive health outcomes among adolescents in the region [29,30].

Compared to FSW who started selling sex as adults, those who started while underage report greater social vulnerabilities and less access to sexual and reproductive health services. Underage initiation of selling sex has been associated with lower education among FSW in Burkina Faso, Cameroon, Eswatini, and Kenya [22,23,25]. In Côte d’Ivoire and Lesotho, experiencing forced sex before the age of 18 has been more commonly reported among FSW who started selling sex while underage than among FSW who started as adults [23,25]. FSW who started selling sex while underage had higher odds of giving birth before age 18 in Côte d’Ivoire and having an unwanted pregnancy in Burkina Faso and the Gambia than FSW who entered sex work as adults [22,23,28]. In contrast, compared to FSW who started as adults, FSW who started selling sex while underage had lower odds of having any children in Eswatini [31] and Kenya [32] and had fewer children on average in Ethiopia [33]. Sexually transmitted infection (STI) symptoms were more commonly reported among FSW who started selling sex while underage in Cameroon and Mozambique compared to FSW who sold sex only as adults [21,34].

Despite these elevated risks, underage entry into selling sex was associated with lower uptake of HIV and STI testing in Burkina Faso, Cameroon, and Mozambique [21,22,34]. Barriers to condom use have been associated with underage entry into selling sex among FSW in Burkina Faso, Cameroon, Kenya, Lesotho, and Mozambique [21,23,26,28,34]. In a study with women and girls aged 14 and older selling sex in Uganda, younger age was associated with lower odds of dual use of condoms and non-barrier contraception [19]. While qualitative research has suggested that contraceptive use, prenatal care, and birthing environments of FSW who started selling sex as minors require further examination [35], there have been no published quantitative studies on whether selling sex as a minor has a long-term effect on use of types of contraception other than condoms and uptake of antenatal care services among those who become pregnant.

Although studies have been conducted with underage girls who sell sex in Angola [20] and Mozambique [21], there is a dearth of research on the long-term health consequences of underage entry into selling sex among FSW in countries in Africa previously colonized by Portugal. Guinea-Bissau is a West African country that declared independence from Portugal in 1973 and is bordered by Senegal and Guinea. The Gross Domestic Product per capita of Guinea-Bissau is $812.96 United States dollars [36]. Adolescents aged 10–19 make up 23% of the population [37]. The primary school completion rate for girls is 57.28% [38]. In 2019, 11% of girls aged 10–17 years old were engaged in economic activity exceeding the threshold for child labor [37]. In Guinea-Bissau, sex work is not legally specified (i.e., neither criminalized nor legalized) [39]. About 2 million people live in all eight regions of Guinea-Bissau, and the population size estimate of sex workers in four regions of the country (Bissau, Bissorã, Bafatá, and Gabú) is 7900 [40]. In one study, over one quarter of FSW in Guinea-Bissau were living with HIV, compared to about 3.8% among all reproductive-age women [41]. In the same study, FSW under the age of 30 had higher odds of undiagnosed HIV infection than those aged 30 or older. The objective of this research is to examine the prevalence of and factors associated with underage initiation of selling sex among adult FSW in Guinea-Bissau. This study aims to compare the sociodemographics, sexual and reproductive health risk factors, and uptake of health services among FSW who started selling sex while underage to those who started as adults.

## 2. Materials and Methods

### 2.1. Study Design

FSW were recruited for this cross-sectional observational study through respondent-driven sampling (RDS) in Bafatá, Bissau, Bissorã, and Gabu in Guinea-Bissau from September through November 2017 [42,43]. RDS is a peer-recruitment method to reach hidden populations [44]. The study team in collaboration with community representatives selected 17 initial “seeds” who completed study procedures and were invited to refer three of their peers to participate. This process was repeated until the target sample size was met. The sample size calculation was powered to estimate HIV prevalence at each site.

### 2.2. Inclusion/Exclusion Criteria

To be eligible, participants needed to be at least 18 years old, a resident in one of the study cities, capable of providing informed consent, and self-report that their sex assigned at birth was female and they earned over half of their income in the past 12 months from selling sex.

### 2.3. Procedure and Data Collection

All eligible participants gave informed consent and completed a questionnaire administered by a trained interviewer in a private location. The questionnaire was based on a modified social ecological model to assess individual-, community-, network- and structural-level HIV risks among FSW [45]. Data collection was approved by the National Research Ethics Committee of Guinea-Bissau. The Johns Hopkins Bloomberg School of Public Health Institutional Review Board approved secondary data analysis of de-identified data.

### 2.4. Data Analysis

In this secondary analysis of the study data, FSW who reported selling sex before the age of 18 were categorized as having initiated selling sex while underage and compared to those who started at 18 years or older using χ^2^ statistics for categorical variables and *t*-tests for continuous variables. Bivariate logistic regression analyses were then conducted using underage entry into selling sex as the dependent variable. Variables significantly related to initiation of selling sex while underage in bivariate models (*p* < 0.05) were considered for inclusion in a multivariable logistic regression model. The variables in the final model were chosen based on the lowest Akaike information criterion through a forward selection method using the swaic command in StataMP 16 (College Station, TX, USA). RDS-adjustment was not conducted because data were pooled from all four cities, and the participants do not represent a single network of FSW.

### 2.5. Measures

Independent variables were chosen based on factors found to be associated with underage initiation of selling sex in prior studies. Age at the time of study was a continuous independent variable [23]. Number of biological children and income were ordinal independent variables [33]. Dichotomous independent variables included literacy [33] in Portuguese and/or Creole; education (primary school or higher); ever experiencing forced sex while underage [23,25]; ever being pregnant; receiving antenatal care during their most recent pregnancy; wanting children or more children in the future at the time of the study; use of emergency contraception ever; current contraceptive method(s) (sterilization, pill, implant, intrauterine device [IUD], injectable, male condoms, female condoms [21]); experiencing STI symptoms in the past 12 months [34]; having ever been tested for HIV [18,22]; and knowledge that anal sex carries the highest risk for HIV acquisition [21]. Non-ordinal categorical independent variables included city, marital status [6] (single and never married, stable partner that is not a spouse, married, divorced/separated, or widowed), and employment status other than sex work (unemployed, employed, or student).

## 3. Results

As shown in Table 1, the mean age of FSW in Guinea-Bissau was 25.2 (standard deviation [SD] = 6.2, range 18–50). Nearly all participants were born in Guinea-Bissau (94.3%, 476/505). Most resided in the city of Bissau (60.6%, 306/505). Only 58.7% (296/504) had completed primary school education or higher. Over one quarter (27.5%, 138/502) could not read or write. Over three quarters were single and never married (79.2%, 400/505). Over half (54.0%, 272/504) were unemployed other than selling sex. Most (70.3%, 348/495) earned 50,000 XOF (76 United States dollars) per month or less. A total of 4.0% (20/501) were forced to have sex when under the age of 18. Over half (58.5%, 289/494) had ever been pregnant. Of those, 71.5% (213/298) received antenatal care during their last pregnancy. Overall, 40.4% (201/498) had no biological children, 27.7% (138/498) had one child, and 31.9% (159/498) had more than one child. About two thirds (66.1%, 334/505) said they plan or hope to have children or more children in the future. Overall, 81.6% (411/504) were using some form of contraception at the time of the study. STI symptoms in the past year were reported by 29.7% (144/485). Less than one third (30.1%, 151/502) had ever tested for HIV. Only 13.1% (66/504) knew that anal sex put them at higher risk of HIV infection than vaginal or oral sex. Over one quarter of FSW (26.3%, 133/505) started selling sex before the age of 18.

The mean current age of FSW who started selling sex while underage was younger than that of FSW who did not (22.0 vs. 26.4 years). A higher percentage of women who started selling sex while underage were born in Guinea-Bissau (97.7% vs. 93.0%) and literate (79.7% vs. 69.9%) than women who did not. Compared to participants who did not report selling sex while underage, a higher proportion of participants who did were single and never married or in a relationship with a stable partner, while a lower proportion were married, divorced, separated, or widowed. A higher percentage of those who started selling sex while underage were students or had a job outside of sex work than those who did not. About half of participants who started selling sex while underage earned 50,000 XOF or less per month compared to about three quarters of participants who started selling sex as adults (51.9% vs. 76.9%). A higher proportion of women who started selling sex while underage were forced to have sex as a minor than women who did not (9.0% vs. 2.2%). Participants who started selling sex while underage were more likely to have no biological children (57.4% vs. 34.4%) and less likely to have one child (24.0% vs. 29.0%) or more than one child (18.6% vs. 36.6%) than participants who did not. A lower percentage of those who started selling sex as minors had ever been pregnant (38.8% vs. 65.5%), received antenatal care (56.6% vs. 74.7%), and wanted more children (51.1% vs. 71.5%). Although more FSW who started selling sex while underage used any form of contraception than FSW who did not (87.9% vs. 79.3%), fewer used most of the contraceptive methods investigated in the study: emergency contraception (6.9% vs. 16.6%), sterilization (4.6% vs. 15.8%), implant (29.5% vs. 42.5%), IUD (7.6% vs. 8.1%), injectable (0.8% vs. 4.0%), and female condoms (3.0% vs. 6.5%). In contrast, a higher proportion of women who started selling sex while underage used oral contraceptives (10.6% vs. 5.1%) or male condoms (51.5% vs. 36.3%). Participants who started selling sex while underage had a higher prevalence of STI symptoms in the past 12 months (38.0% vs. 26.7%). Fewer of those who started selling sex while underage had ever been tested for HIV (16.5% vs. 35.0%). Knowledge that anal sex carries higher risk for HIV acquisition was less prevalent among FSW who started selling sex while underage compared to those who did not (3.0% vs. 16.7%).

As shown in Table 2, underage entry into selling sex was associated with younger age at the time of the study (odds ratio (OR): 0.84; 95% confidence interval (CI): 0.80, 0.89). Compared to women who started selling sex as adults, women who started selling sex while underage had lower odds of living in Bafatá (OR: 0.54; 95% CI: 0.31, 0.95) vs. Bissau. Underage initiation of selling sex was associated with being literate in Portuguese and/or Creole (OR: 1.69; 95% CI: 1.05, 2.72). FSW who started selling sex while underage had lower odds of being married (OR: 0.21; 95% CI: 0.06, 0.71) or widowed (OR: 0.12; 95% CI: 0.02, 0.91) vs. single. Underage entry into selling sex was associated with having an additional job outside of selling sex (OR: 1.86; 95% CI: 1.15, 3.00) or being a student (OR: 2.17; 95% CI: 1.33, 3.56) vs. having no other job. Entry into selling sex while underage was associated with higher income (OR: 1.39; 95% CI: 1.24, 1.56). Those who started selling sex while underage had over four times the odds of having been forced to have sex as a minor (OR: 4.46; 95% CI: 1.78, 11.18). Participants who started selling sex while underage had lower odds of ever being pregnant (OR: 0.33; 95% CI: 0.22, 0.51), receiving antenatal care while pregnant (OR: 0.44; 95% CI: 0.24, 0.82), or wanting any children or more children (OR: 0.42; 95% CI: 0.28, 0.63). Underage initiation of selling sex was associated with having fewer biological children (OR: 0.55, 0.42, 0.70). Women who started selling sex while underage had lower odds of ever using emergency contraception (OR: 0.37; 95% CI: 0.18, 0.77), being sterilized (OR: 0.26; 95% CI: 0.11, 0.61), or currently using a contraceptive implant (OR: 0.57; 95% CI: 0.37, 0.87) but higher odds of using any contraception (OR: 1.89; 95% CI: 1.06, 3.38), oral contraception (OR: 2.20; 95% CI: 1.07, 4.53) or male condoms for contraception (OR: 1.87; 95% CI: 1.25, 2.79). Underage entry into selling sex was associated having STI symptoms (OR: 1.68; 95% CI: 1.10, 2.58), never being tested for HIV (OR: 0.37; 95% CI: 0.22, 0.61), and not knowing that anal sex puts them at most risk for HIV infection (OR: 0.15; 95% CI: 0.06, 0.43). Country of birth, education, and using an IUD, injectable contraception, or female condoms were not significantly related to underage initiation into selling sex.

In the multivariable analysis, as shown in Table 3, the relationship between underage entry into selling sex and younger age remained statistically significant (adjusted odds ratio [aOR]: 0.82; 95% CI: 0.76, 0.89). FSW who started selling sex while underage had higher odds of living in Gabu vs. Bissau (OR: 6.17; 95% CI: 2.35, 16.15) and having a stable partner who is not a spouse (OR: 8.70; 95% CI: 2.63, 28.79). Participants who started selling sex while underage had over six times the odds of ever experiencing forced sex as a minor (aOR: 6.74; 95% CI: 2.05, 22.13). Despite having lower odds of ever being pregnant (aOR: 0.40; 95% CI: 0.22, 0.73), those who started selling sex while underage had lower odds of saying they plan to or hope to have children or more children in the future (aOR: 0.31; 95% CI: 0.17, 0.56) and lower odds of reporting using a contraceptive implant (aOR: 0.43; 95% CI: 0.24, 0.77). Initiation of selling sex while underage was inversely associated with ever being tested for HIV (aOR: 0.43; 95% CI: 0.20, 0.91) and knowledge that anal sex is most risky for HIV acquisition (OR: 0.12; 95% CI: 0.02, 0.59). Experiencing STI symptoms in the past year was not significantly related to underage entry into selling sex in the multivariable model.

## 4. Discussion

In this study more than one in four FSW started selling sex while underage. FSW who started selling sex while underage had higher odds of experiencing forced sex before the age of 18 and lower odds of ever being pregnant, wanting to have children, using contraceptive implants, ever testing for HIV, and knowing that anal sex puts them most at risk for HIV than FSW who started selling sex as adults.

Similar to other studies with FSW in Africa [23,25], underage entry into selling sex in Guinea-Bissau was also associated with experiencing forced sex while underage. This would indicate a potential need for health services such as emergency contraception and HIV testing, but in this sample among those who started selling sex while underage, less than 7% and 17% had ever used these services, respectively [46].

Prior research in other African countries has shown that underage entry into selling sex was associated with having an unwanted pregnancy [22,26] and giving birth while underage [25]. However, in Guinea-Bissau women who started selling sex while underage had lower odds of reporting ever being pregnant and had fewer biological children than women who started selling sex as adults. Despite reported preferences to not have children or more children, FSW in this study who started selling sex while underage had lower odds than those who started as adults of using highly effective contraceptive methods such as implants [47]. Access to a range of highly effective contraceptive methods is particularly important for women who started selling sex while underage who do not want to have children because pregnancy and childbirth can have serious negative physical and psychological health consequences for women with a history of childhood commercial sexual exploitation or sexual abuse [48,49].

Receiving antenatal care is associated with better maternal and infant health outcomes [50], but over forty percent of FSW who started selling sex while underage in Guinea-Bissau who had ever been pregnant did not receive antenatal care. This is much higher than the 3% of all women aged 15–49 in Guinea-Bissau who did not receive antenatal care [51]. Destigmatizing and making antenatal care services friendly toward adolescents who sell sex may improve uptake among this population [52].

Over half of FSW in this study who started selling sex while underage reported current use of male condoms for contraception, which can also help protect them from HIV and other STIs. However, over one third had experienced STI symptoms in the past 12 months, suggesting the need for further prevention and treatment services. Data on the prevalence of STI symptoms in Guinea-Bissau are limited, but the prevalence among FSW was about ten times higher than the prevalence of symptoms among women aged 16–40 on Bubaque Island, 3.3% [53].

In this study, only 3% of FSW who started selling sex while underage knew that anal sex carries the highest risk for HIV acquisition rather than vaginal or oral sex. This is notable because other research in Burkina Faso has shown that FSW who started selling sex while underage were more likely to have had anal sex in the past year [22]. This gap in knowledge about HIV may be due to limited access to information. In other research, FSW underage entry into selling sex was associated with not attending talks or meetings about HIV [22]. Updating programs used for comprehensive sex education to include the specifics of transmission risks could help address this limited knowledge among adolescents who sell sex and FSW who started selling sex as minors.

This study has some limitations. This is a secondary data analysis, and the sample size may not have been large enough to detect differences in some indicators between those who started selling sex while underage and those who did not. A larger sample size could for example lead to more significant differences in use of other contraceptive methods. The data are cross-sectional, therefore causal inferences could not be made. Adjustment for the RDS design was not possible due to pooling data from multiple cities. The sample only included adult FSW, so inferences cannot be made about girls who sell sex while underage and stop selling sex before adulthood. Data on many indicators in the study, such as being forced to have sex while underage, are not available for all women in Guinea-Bissau, so they cannot be compared to the FSW in this study [54]. Additionally, the data are self-reported and may be subject to inaccurate recall and social desirability bias. Future studies could use self-administered surveys to reduce the possibility of social desirability bias.

## 5. Conclusions

Despite these limitations, this study contributes to the limited research on sex work in Guinea-Bissau [41]. Given the high prevalence of underage entry into selling sex and the suboptimal uptake of sexual and reproductive health services among this group, these findings underscore the need to prevent early initiation of selling sex and focus programmatic attention on overcoming barriers to care among those who previously sold or currently sell sex while underage. A combination prevention approach emphasizing structural interventions is needed to address HIV, STIs, unintended pregnancies, and negative maternal and child health outcomes among those who start selling sex while underage in Guinea-Bissau.

## Figures and Tables

**Table 1 ijerph-19-12715-t001:** Characteristics of female sex worker study participants (n = 505) in Guinea-Bissau by early or later initiation of selling sex, 2017.

Variable	Started Selling Sex 18+	Started Selling Sex <18	Total	*p*-Value ^1^
**Current age [mean (SD)]**	26.4 (6.5)	22.0 (4.1)	25.2 (6.2)	<0.001
**Born in Guinea-Bissau**	93.0% (346/372)	97.7% (130/133)	94.3% (476/505)	0.044
**City**				0.047
Bissau	59.7% (222/372)	63.2% (84/133)	60.6% (306/505)	
Bissora	8.9% (33/372)	10.5% (14/133)	9.3% (47/505)	
Bafata	23.7% (88/372)	13.5% (18/133)	21.0% (106/505)	
Gabu	7.8% (29/372)	12.8% (17/133)	9.1% (46/505)	
**Completed primary school or higher ^2^**	57.1% (212/371)	63.2% (84/133)	58.7% (296/504)	0.227
**Literate in Portuguese and/or Creole ^3^**	69.9% (258/369)	79.7% (106/133)	72.5% (364/502)	0.030
**Current relationship status**				0.001
Single and never married	76.1% (283/372)	88.0% (117/133)	79.2% (400/505)	
Stable partner that is not spouse	3.5% (13/372)	6.8% (9/133)	4.4% (22/505)	
Married	9.1% (34/372)	2.3% (3/133)	7.3% (37/505)	
Divorced/Separated	5.9% (22/372)	2.3% (3/133)	5.0% (25/505)	
Widowed	5.4% (20/372)	0.8% (1/133)	4.2% (21/505)	
**Other work ^2^**				0.003
No additional work	58.5% (217/371)	41.4% (55/133)	54.0% (272/504)	
Outside work	22.9% (85/371)	30.1% (40/133)	24.8% (125/504)	
Student	18.6% (69/371)	28.6% (38/133)	21.2% (107/504)	
**Average monthly income in West African CFA Francs (XOF) ^4^**				<0.001
0	52.7% (192/364)	26.0% (34/131)	45.7% (226/495)	
1–50,000	24.2% (88/364)	26.0% (34/131)	24.6% (122/495)	
50,001–100,000	9.1% (33/364)	10.7% (14/131)	9.5% (47/495)	
100,001–150,000	3.8% (14/364)	6.1% (8/131)	4.4% (22/495)	
150,001–200,000	3.3% (12/364)	21.4% (28/131)	8.1% (40/495)	
200,001–250,000	4.9% (18/364)	6.9% (9/131)	5.5% (27/495)	
>250,000	1.9% (7/364)	3.1% (4/131)	2.2% (11/495)	
**Was forced to have sex <18 years old ^5^**	2.2% (8/368)	9.0% (12/133)	4.0% (20/501)	0.001
**Ever pregnant ^6^**	65.5% (239/365)	38.8% (50/129)	58.5% (289/494)	<0.001
**Received antenatal care ^7^**	74.7% (183/245)	56.6% (30/53)	71.5% (213/298)	0.008
**Ever had an abortion**	4.3% (16/369)	2.3% (3/131)	3.8% (19/500)	0.293
**Number of biological children ^8^**				<0.001
None	34.4% (127/369)	57.4% (74/129)	40.4% (201/498)	
One	29.0% (107/369)	24.0% (31/129)	27.7% (138/498)	
Two or more	36.6% (135/369)	18.6% (24/129)	31.9% (159/498)	
**Wants (more) children**	71.5% (266/372)	51.1% (68/133)	66.1% (334/505)	<0.001
**Ever used emergency contraception ^9^**	16.6% (61/367)	6.9% (9/130)	14.1% (70/497)	0.006
**Sterilized ^10^**	15.8% (58/366)	4.6% (6/130)	12.9% (64/496)	0.001
**Contraceptive method currently using ^2^**				
Pill	5.1% (19/372)	10.6% (14/132)	6.5% (33/504)	0.028
Implant	42.5% (158/372)	29.5% (39/132)	39.1% (197/504)	0.009
IUD	8.1% (30/372)	7.6% (10/132)	7.9% (40/504)	0.858
Injectable	4.0% (15/372)	0.8% (1/132)	3.2% (16/504)	0.065
Male condom	36.3% (135/372)	51.5% (68/132)	40.3% (203/504)	0.002
Female condom	6.5% (24/372)	3.0% (4/132)	5.6% (28/504)	0.140
Any form of birth control	79.3% (295/372)	87.9% (116/132)	81.5% (411/504)	0.029
**STI symptoms past 12 months ^11^**	26.7% (95/356)	38.0% (49/129)	29.7% (144/485)	0.016
**Ever tested for HIV ^12^**	35.0% (129/369)	16.5% (22/133)	30.1% (151/502)	<0.001
**Knowledge that anal sex puts you most at risk for HIV ^2^**	16.7% (62/371)	3.0% (4/133)	13.1% (66/504)	<0.001

SD = standard deviation, HIV = human immunodeficiency virus, IUD = intrauterine device, STI = sexually transmitted infection. ^1^
*t*-tests were conducted for continuous variables. χ^2^ tests were conducted for categorical variables. ^2^ n = 1 missing. ^3^ n = 3 missing. ^4^ n = 10 missing. ^5^ n = 4 missing. ^6^ n = 11 missing. ^7^ Among those who were ever pregnant. ^8^ n = 7 missing. ^9^ n = 8 missing. ^10^ n = 9 missing. ^11^ n = 20 missing. ^12^ n = 3 missing. Bold is to indicate a variable name.

**Table 2 ijerph-19-12715-t002:** Bivariate logistic regression analyses of correlates of underage initiation of selling sex among female sex worker study participants (n = 505) in Guinea-Bissau, 2017.

Variable	Odds Ratio	95% Confidence Interval	*p*-Value
**Current age (continuous)**	0.84	0.80–0.89	<0.001
**Born in Guinea-Bissau ^1^**	3.26	0.97–10.94	0.056
**City**			
Bissau (ref)			
Bissora	1.12	0.57–2.20	0.739
Bafata	0.54	0.31–0.95	0.033
Gabu	1.55	0.81–2.97	0.186
**Completed primary school or higher ^1^**	1.29	0.86–1.93	0.227
**Literate in Portuguese and/or Creole ^1^**	1.69	1.05–2.72	0.031
**Current relationship status**			
Single and never married (ref)			
Stable partner that is not spouse	1.67	0.70–4.02	0.249
Married	0.21	0.06–0.71	0.012
Divorced/Separated	0.33	0.10–1.12	0.076
Widowed	0.12	0.02–0.91	0.040
**Other work**			
No additional work (ref)			
Outside work	1.86	1.15–3.00	0.011
Student	2.17	1.33–3.56	0.002
**Average monthly income in West African CFA Francs (XOF) (ordinal)**	1.39	1.24–1.56	<0.001
**Was forced to have sex <18 years old ^1^**	4.46	1.78–11.18	0.001
**Ever pregnant ^1^**	0.33	0.22–0.51	<0.001
**Received antenatal care ^1^**	0.44	0.24–0.82	0.009
**Ever had an abortion ^1^**	0.52	0.15–1.80	0.301
**Number of biological children (ordinal)**	0.55	0.42–0.70	<0.001
**Wants (more) children ^1^**	0.42	0.28–0.63	<0.001
**Ever used emergency contraception ^1^**	0.37	0.18–0.77	0.008
**Sterilized ^1^**	0.26	0.11–0.61	0.002
**Contraceptive method currently using ^1^**			
Pill	2.20	1.07–4.53	0.032
Implant	0.57	0.37–0.87	0.009
IUD	0.93	0.44–1.97	0.858
Injectable	0.18	0.02–1.39	0.100
Male condom	1.87	1.25–2.79	0.002
Female condom	0.45	0.15–1.33	0.150
Any form of birth control	1.89	1.06–3.38	0.031
**STI symptoms past 12 months ^1^**	1.68	1.10–2.58	0.017
**Ever tested for HIV ^1^**	0.37	0.22–0.61	<0.001
**Knowledge that anal sex puts you most at risk for HIV ^1^**	0.15	0.06–0.43	<0.001

HIV = human immunodeficiency virus, IUD = intrauterine device, STI = sexually transmitted infection. ^1^ For these variables, the reference category is “no” and the category shown in the table is “yes”. Bold is to indicate a variable name.

**Table 3 ijerph-19-12715-t003:** Multivariable logistic regression analysis of correlates of underage initiation of selling sex among female sex worker study participants (n = 468) in Guinea-Bissau, 2017.

Variable	Adjusted Odds Ratio	95% Confidence Interval	*p*-Value
**Current age (continuous)**	0.82	0.76–0.89	<0.001
**City**			
Bissau (ref)			
Bissora	0.88	0.37–2.09	0.767
Bafata	0.78	0.36–1.69	0.533
Gabu	6.17	2.35–16.15	<0.001
**Current relationship status**			
Single and never married (ref)			
Stable partner who is not spouse	8.70	2.63–28.79	<0.001
Married	0.69	0.17–2.77	0.603
Divorced/Separated	0.70	0.09–5.37	0.731
Widowed	0.31	0.03–3.18	0.327
**Was forced to have sex <18 years old ^1^**	6.74	2.05–22.13	0.002
**Ever pregnant ^1^**	0.40	0.22–0.73	0.003
**Wants (more) children ^1^**	0.31	0.17–0.56	<0.001
**Currently using contraceptive implant ^1^**	0.43	0.24–0.77	0.004
**STI symptoms past 12 months ^1^**	1.66	0.95–2.90	0.077
**Ever tested for HIV ^1^**	0.43	0.20–0.91	0.027
**Knowledge that anal sex puts you most at risk for HIV ^1^**	0.12	0.02–0.59	0.009

HIV = human immunodeficiency virus, STI = sexually transmitted infection. All variables included in the multivariate model are shown in the table. ^1^ For these variables, the reference category is “no” and the category shown in the table is “yes”. Bold is to indicate a variable name.

## Data Availability

The data that support the findings of this study are available from the authors with the permission of the Guinea-Bissau Ministry of Public Health.

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
