# Peer review of "Sexual and Reproductive Health Service Needs Associated with Underage Initiation of Selling Sex among Adult Female Sex Workers in Guinea-Bissau"

_ijerph, 2022, doi:10.3390/ijerph191912715_

Round 1

Reviewer 1 Report

1. You must take care of the space. It also frequently occurs after a full stop

2. Line 71: This introduction only talks about data in relation to sexual and reproductive health among underage sex workers. However, this objective involves much more complex issues. The introduction of the article completely forgets to mention and describe the socioeconomic context that houses the phenomenon that it tries to approach. As well as the common and divergent aspects that relate Guinea-Bissau with the rest of the countries mentioned.

3. Line 121: It would be VERY IMPORTANT to compare these data with those of the female population in general in the country, to see if the prevalence of these phenomena is higher in sex workers than in other women of similar ages.

4. Line 133: For example: Is this percentage higher or lower among sex workers than among all women in the country?

5. In order to be able to affirm this with such force, it is necessary to compare percentages of these social facts (or independent variables) with the rest of the female population and to talk about the social context in terms of gender differences and sexual and reproductive health in the country. Only with the data obtained with a reduced sample in a survey and quantitative analysis process (without collecting the discourse or stories of the interviewees) this cannot be scientifically concluded. In any case, it can be considered as an emerging hypothesis to be tested in a more holistic future study.

Author Response

We appreciate the reviewer’s helpful suggestions. Our responses to each point are below in bold:

  1. "You must take care of the space. It also frequently occurs after a full stop."

We adjusted the spacing before the references and punctuation.

  1. Line 71: This introduction only talks about data in relation to sexual and reproductive health among underage sex workers. However, this objective involves much more complex issues. The introduction of the article completely forgets to mention and describe the socioeconomic context that houses the phenomenon that it tries to approach. As well as the common and divergent aspects that relate Guinea-Bissau with the rest of the countries mentioned.

We added the following to the introduction:

“Underage initiation of selling sex has been associated with lower education among FSW in Burkina Faso, Cameroon, Eswatini, and Kenya.”

“The Gross Domestic Product per capita of Guinea-Bissau is $812.96 United States dollars. Adolescents aged 10-19 make up 23% of the population. The primary school completion rate for girls is 57.28% [38]. In 2019, 11% of girls aged 10-17 years old were engaged in economic activity exceeding the threshold for child labor.”

  1. Line 121: It would be VERY IMPORTANT to compare these data with those of the female population in general in the country, to see if the prevalence of these phenomena is higher in sex workers than in other women of similar ages.

We added the following to the discussion section:

“This is much higher than the 3% of all women aged 15-49 in Guinea-Bissau who did not receive antenatal care.”

“Data on the prevalence of STI symptoms in Guinea-Bissau are limited, but the prevalence among FSW was about ten times higher than the prevalence of symptoms among women aged 16-40 on Bubaque Island, 3.3%.”

  1. Line 133: For example: Is this percentage higher or lower among sex workers than among all women in the country?

We added the following to the limitations section:

“Data on many indicators in the study such as being forced to have sex while underage are not available for all women in Guinea-Bissau, so they cannot be compared to the FSW in this study.”

  1. In order to be able to affirm this with such force, it is necessary to compare percentages of these social facts (or independent variables) with the rest of the female population and to talk about the social context in terms of gender differences and sexual and reproductive health in the country. Only with the data obtained with a reduced sample in a survey and quantitative analysis process (without collecting the discourse or stories of the interviewees) this cannot be scientifically concluded. In any case, it can be considered as an emerging hypothesis to be tested in a more holistic future study.

The purpose of this study is to compare female sex workers who started selling sex as adults to those who started while underage, not to compare female sex workers to the rest of the population. We have clarified that in the language in the discussion.

Reviewer 2 Report

Given the high prevalence of HIV infection (key population) this study will contribute to public and collective health, but also to redefine health policies.

The objective of this research is to examine the prevalence of and factors associated with underage initiation of selling sex among adult FSW in Guinea-Bissau.

I have some improvement suggestions to consider by the authors:

1. I suggest that in the introduction you include a paragraph on current health publicity in the context of hiv infection and other STIs. WHO - 2030 Agenda - ODS 3.

2. Materials and Methods - recruitment should be better explained;

3. " completed an interviewer-administered questionnaire" - how did you ensure that the results were not biased?

4. They should present how the instrument was developed. ex. bibliographic research, expertise, etc....

5. In conclusion, you can talk about Combined Prevention, emphasizing the structural dimension.

Congratulations to the authors. Innovative work.

Author Response

We appreciate the reviewer’s helpful suggestions. Our responses to each point are below in bold:

Given the high prevalence of HIV infection (key population) this study will contribute to public and collective health, but also to redefine health policies.

The objective of this research is to examine the prevalence of and factors associated with underage initiation of selling sex among adult FSW in Guinea-Bissau.

I have some improvement suggestions to consider by the authors:

Thank you.

  1. I suggest that in the introduction you include a paragraph on current health publicity in the context of hiv infection and other STIs. WHO - 2030 Agenda - ODS 3.

We added the following to the introduction: “The third Sustainable Development Goal on good health and well-being includes targets to end the AIDS epidemic and ensure universal access to sexual and reproductive healthcare services including family planning, information and education by 2030. The fifth Sustainable Development Goal on gender equality also includes a target to ensure universal access to sexual and reproductive health and reproductive rights. Identifying and addressing the needs of vulnerable populations are important components of achieving these targets.”

  1. Materials and Methods - recruitment should be better explained;

We added the following to the methods section: “RDS is a peer-recruitment method to reach hidden populations. The study team in collaboration with community representatives selected 17 initial “seeds” who completed study procedures and were invited to refer three of their peers to participate. This process was repeated until the target sample size was met.”

  1. " completed an interviewer-administered questionnaire" - how did you ensure that the results were not biased?

We added to the methods section that questionnaires were conducted in a private location by trained interviewers to reduce the risk of biased results.

  1. They should present how the instrument was developed. ex. bibliographic research, expertise, etc....

We added the following to the methods section: “The questionnaire was based on a modified social ecological model to assess individual-, community-, network- and structural-level HIV risks among FSW.”

  1. In conclusion, you can talk about Combined Prevention, emphasizing the structural dimension.

We edited the conclusion to state: “A combination prevention approach emphasizing structural interventions is needed to address HIV, STIs, unintended pregnancies, and negative maternal and child health out-comes among those who start selling sex while underage in Guinea-Bissau.”

Congratulations to the authors. Innovative work.

Thank you.

Reviewer 3 Report

The paper entitled “Sexual and reproductive health service needs associated with underage initiation of selling sex among adult female sex workers in Guinea-Bissau” focuses on an important issue relating to public health.

The article seems interesting; however, I have some considerations:

1. Was the effect and sample size calculated? Please indicate this in the methods.

2. The Materials and method section should be divided into subsections maintaining the structure (i.e. Study design; Inclusion/exclusion criteria; Procedure and data collection; Measure; Data analysis) - in its current form, it has a chaotic structure, which makes it difficult to read

3. RDS should be at least briefly explained

4. Please show the number of children as a categorical variable. The statistical program will count it all, but what does it mean that they had an average of 1.2 children? I recommend defining, for example, three categories: women who do not have children, who have one child and two or more. Only the variable in this form should be included in the analysis.

5. Concerning Table 1 and missing data information below the table - please reformulate the annotation - you can apply superscript to specific variables and explain it in the footer below the table. It is hardly legible in its present form and even incomprehensible to an inexperienced reader.

6. Please expand the Limitations section with indications for further research. For example, if the authors write that the sample size may be insufficient to detect differences in some indicators - please mention which indicators may be particularly important in the authors' opinion. If the limitation is self-report data collection and the resulting consequences - how to design further studies to reduce the risk of inaccurate recall and social desirability bias.

Author Response

We appreciate the reviewer’s helpful suggestions. Our responses to each point are below in bold:

The paper entitled “Sexual and reproductive health service needs associated with underage initiation of selling sex among adult female sex workers in Guinea-Bissau” focuses on an important issue relating to public health. The article seems interesting; however, I have some considerations:

Thank you.

  1. Was the effect and sample size calculated? Please indicate this in the methods.

We added the following to the methods section: “The sample size calculation was powered to estimate HIV prevalence at each site.” 

  1. The Materials and method section should be divided into subsections maintaining the structure (i.e. Study design; Inclusion/exclusion criteria; Procedure and data collection; Measure; Data analysis) - in its current form, it has a chaotic structure, which makes it difficult to read

We divided the section into subsections as suggested.

  1. RDS should be at least briefly explained

We added the following to the methods section: “RDS is a peer-recruitment method to reach hidden populations. The study team in collaboration with community representatives selected 17 initial “seeds” who completed study procedures and were invited to refer three of their peers to participate. This process was repeated until the target sample size was met.”

  1. Please show the number of children as a categorical variable. The statistical program will count it all, but what does it mean that they had an average of 1.2 children? I recommend defining, for example, three categories: women who do not have children, who have one child and two or more. Only the variable in this form should be included in the analysis.

We categorized the number of children as suggested.

  1. Concerning Table 1 and missing data information below the table - please reformulate the annotation - you can apply superscript to specific variables and explain it in the footer below the table. It is hardly legible in its present form and even incomprehensible to an inexperienced reader.

We reformatted the annotation as suggested.

  1. Please expand the Limitations section with indications for further research. For example, if the authors write that the sample size may be insufficient to detect differences in some indicators - please mention which indicators may be particularly important in the authors' opinion. If the limitation is self-report data collection and the resulting consequences - how to design further studies to reduce the risk of inaccurate recall and social desirability bias.

We added the following to the limitations section: “A larger sample size could for example lead to more significant differences in use of other contraceptive methods…Future studies could use self-administered surveys to reduce the possibility of social desirability bias.”

Reviewer 4 Report

This is very well written paper and I have just a few suggestions to the authors. 1It would be very informative for the readers to present the sociodemographic profile of the respondents and to add the description of the sample. Information about the data collection needs also to be included in the manuscript since the study is focused on a specific group of respondents that is difficult to reach. The aims of the paper should be also described more explicitly in the text. On Table 2 and 3 you can add a notation for the change of the reference category for a particular group. Instead only text is includes in the presnet version of the manuscript: “If no category is listed, the reference category is “no” and the category shown in the table is “yes”.

Author Response

We appreciate the reviewer’s helpful suggestions. Our responses to each point are below in bold:

This is very well written paper and I have just a few suggestions to the authors.

Thank you.

It would be very informative for the readers to present the sociodemographic profile of the respondents and to add the description of the sample.

We added education and income as sociodemographic variables to describe the sample.

Information about the data collection needs also to be included in the manuscript since the study is focused on a specific group of respondents that is difficult to reach.

We added the following to the methods section: “RDS is a peer-recruitment method to reach hidden populations. The study team in collaboration with community representatives selected 17 initial “seeds” who completed study procedures and were invited to refer three of their peers to participate. This process was repeated until the target sample size was met.”

The aims of the paper should be also described more explicitly in the text.

We added the following to the introduction: “This study aims to compare the sociodemographics, sexual and reproductive health risk factors, and uptake of health services among FSW who started selling sex while underage to those who started as adults.”

On Table 2 and 3 you can add a notation for the change of the reference category for a particular group. Instead only text is includes in the presnet version of the manuscript: “If no category is listed, the reference category is “no” and the category shown in the table is “yes”.

We added notations in these tables as suggested.

Round 2

Reviewer 3 Report

Thank you for providing the suggested changes.